# GPT4LoRA: Optimizing LoRA Combination via MLLM Self-Reflection

## Abstract

Low-Rank Adaptation (LoRA) is extensively used in generative models to enable concept-driven personalization, such as rendering specific characters or adopting unique styles. Although recent approaches have explored LoRA combination to integrate diverse concepts, they often require further fine-tuning or modifications to the generative model's original architecture. To address these limitations, we introduce *GPT4LoRA*, a novel method for LoRA combination that adjusts combination coefficients by leveraging the self-reflection capabilities of multimodal large language models (MLLMs). GPT4LoRA operates through a three-step process—*Generate*, *Feedback*, and *Refine*—without the need for additional training, relying solely on tailored prompts and iterative refinement to enhance performance. This iterative approach ensures more constructive feedback and optimizes the model responses. Experiments on various LoRA model combinations, including both realistic and anime styles, demonstrate that GPT4LoRA achieves superior results compared to existing methods. Additionally, an evaluation framework based on GPT-4o further highlights the clear performance gains offered by GPT4LoRA over standard baselines, showcasing its potential for advancing the field.

## 1 Introduction

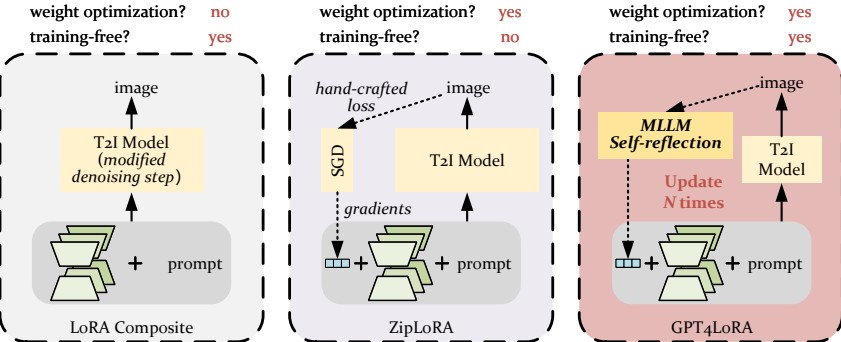

Figure 1: Comparison between GPT4LoRA and some representative LoRA combination methods.

In recent years, advancements in generative modeling techniques have significantly enhanced the ability to produce complex and customized image outputs. Among these developments, Low-Rank Adaptation (LoRA) has emerged as an efficient method for fine-tuning large pre-trained models with minimal computational resources. The flexibility of LoRA in adapting models to distinct attributes and styles has led to its widespread use, particularly in areas where high-quality image generation is critical. However, combining multiple LoRA models to achieve seamless compositions presents a challenge, as current methods often involve complex integration processes that can compromise image quality or demand significant manual adjustments (Ruiz et al., 2023; sce; civ).

Existing approaches to LoRA model combination, such as ZipLoRA (Shah et al., 2023) and LoRA Switch (Zhong et al., 2024), aim to mitigate these difficulties by introducing techniques that modify

coefficient matrices or activate models sequentially during the denoising process. However, these methods often require additional fine-tuning or manual intervention, complicating the workflow and potentially leading to inconsistencies in the final output. While LoRA Composite (Zhong et al., 2024) offers a decoding-centric approach to altering denoising steps, and simpler coefficient adjustment methods have shown some effectiveness (sce), they are computationally costly and impractical when a large number of LoRA models are involved. Furthermore, the absence of robust evaluation mechanisms adds to these challenges, as current approaches rely on manually designed rules or CLIP-based automatic scoring systems, which have been shown to be unreliable in evaluating image quality.

A fundamental limitation of these methods lies in the subjectivity and unreliability of the evaluation process for image quality. Many approaches depend on manually crafted rules or automated evaluators such as CLIP, which often fail to provide consistent and accurate assessments of generated images. This lack of reliable evaluation weakens the effectiveness of LoRA combinations, as the resulting images may not meet the intended quality or adhere to the desired attributes. Consequently, there is a critical need for a more reliable and adaptable approach to optimizing LoRA combinations without reliance on manual designs or unstable scoring mechanisms.

In response to these limitations, we propose GPT4LoRA, a new training-free method for combining LoRA models that leverages the self-reflection capability of multimodal large language models (MLLMs) (Renze & Guven, 2024; Shinn et al., 2024). Unlike previous methods, as shown in Fig. 1 GPT4LoRA generates and refines combination coefficients dynamically, without the need for fine-tuning or modification of the denoising process. By utilizing the self-assessment mechanism of MLLMs, GPT4LoRA provides a more reliable system for evaluating and optimizing LoRA combinations, resulting in higher-quality images with reduced computational overhead. This method operates through an iterative process of generation, feedback, and refinement, enabling continuous improvement of generated images based on real-time evaluations.

Our approach is supported by a carefully designed paradigm for few-shot sample selection, which guides the self-reflection mechanism of the MLLM during the iterative process. GPT4LoRA does not require annotated data or manually designed rules, instead relying on few-shot samples and specifically tailored prompts for generating, evaluating, and refining LoRA combinations. Extensive experiments conducted on a benchmark of widely-used LoRA models demonstrate that GPT4LoRA outperforms existing methods in both quantitative and qualitative evaluations. By eliminating reliance on unreliable automatic scoring systems and harnessing MLLM-based self-reflection, GPT4LoRA establishes a new standard for efficient and high-quality LoRA composition in generative image models.

## 2 RELATED WORK

### 2.1 MODEL MERGING

Using pre-trained models (Rombach et al., 2022; Podell et al., 2023; Liu et al., 2024; Achiam et al., 2023) typically involves fine-tuning them to specialize on a specific task (Devlin, 2018), which can lead to improved performance with a small amount of task-specific labeled data. These benefits have resulted in the release of thousands of fine-tuned checkpoints (Wolf, 2019; civ). However, maintaining a separate fine-tuned model for each task presents challenges: (1) each new task requires storing and deploying a distinct model, and (2) isolated models miss the opportunity to share insights between related tasks, which could boost performance on both similar and new tasks. To solve this problem, a series of model merging techniques (Zhang et al., 2023b; Ilharco et al., 2022; Yadav et al., 2023; Yu et al., 2024) are introduced. Model merging, or model fusion, is a valuable technique that combines the parameters of several distinct models, each with unique capabilities, to create a universal model. This process does not require access to the original training data or involves high computational costs. Although model merging is a relatively young topic, it is evolving rapidly and has already found applications in several domains, such as improving performance on a single target task (Gupta et al., 2020), improving out-of-domain generalization (Jin et al., 2022), compression (Li et al., 2023), multi-modal merging models (Sung et al., 2023), and other settings Don-Yehiya et al. (2022). Recently, the availability of pre-trained and fine-tuned models in the machine-learning community has increased significantly. Open-source platforms such as Huggingface (Wolf, 2019)

provide easy access to a wide range of well-trained models with different capabilities. These comprehensive model repositories facilitate quick advancements in the field of model integration.

## 2.2 LoRA Combination

Recently, diffusion models (Podell et al., 2023; Rombach et al., 2022; Saharia et al., 2022) have allowed for impressive image generation quality with their excellent understanding of diverse artistic concepts and enhanced controllability due to multi-modal conditioning support (with text being the most popular mode). The usability and flexibility of generative models have further progressed with a wide variety of personalization approaches, such as DreamBooth (Ruiz et al., 2023) and StyleDrop (Sohn et al., 2023). These approaches fine-tune a base diffusion model on the images of a specific concept to produce novel renditions in various contexts. Such concepts can be a specific object or person, or an artistic style. Naturally, one may wish to render a specific person in their personal style. To this end, a series of LoRA combination techniques (Yang et al., 2024b; Shah et al., 2023; Zhong et al., 2024) are proposed to fulfill this task. For example, ZipLoRA (Shah et al., 2023) learns mixing coefficients for each column for both style and subject LoRAs and requires a further fine-tuning process to update both mixing coefficients. By utilizing textual, layout, and image-based conditions (optional) to integrate multiple LoRAs, LoRA-Composer (Yang et al., 2024b) alleviates the concept confusion and concept vanishing issues. Instead of directly manipulating the combination coefficients, LoRA Composite (Zhong et al., 2024) concentrates on the denoising process, involving all LoRAs working together as guidance throughout the generation process.

## 2.3 In-Context Learning

In-context learning (ICL) is a recent methodology from natural language processing (NLP), where large models perform tasks they haven't seen before by analyzing a few given examples along with the test instance. This approach is effective because it allows users to adapt the model to various tasks without needing to fine-tune model parameters. Numerous methods have been developed based on in-context learning for tasks such as text classification (Zhang et al., 2022) and machine translation (Zhang et al., 2023a). In the realm of multi-modality learning, in-context learning is still a relatively new concept. Most existing work in this area has focused on employing large image-to-image models for tasks like image inpainting (Bar et al., 2022).

## 2.4 Self-Reflection in LLMs

Self-reflection is a process in which a person thinks about their thoughts, feelings, and behaviors. Similar to humans, this ability allows LLMs to identify errors, explain the cause of these errors, and generate advice to avoid making similar types of errors in the future (Pan et al., 2023; Madaan et al., 2024; Shinn et al., 2024). Reflexion Shinn et al. (2024) converts binary or scalar feedback from the environment into verbal feedback in the form of a textual summary, which is then added as additional context for the LLM agent in the next episode. Self-refine (Madaan et al., 2024) introduces an iterative self-refinement algorithm that alternates between two generative steps, which work in tandem to generate high-quality outputs. In this paper, we follow the philosophy of self-reflection and, for the first time, employ self-reflection and in-context learning ability in MLLMs to LoRA combination.

## 3 Method

### 3.1 Background

**Diffusion Models**

Diffusion models (Rombach et al., 2022) are generative models that create data samples from Gaussian noise via a sequential denoising process. These models utilize a series of denoising autoencoders to estimate the score of a data distribution. The denoising process introduces noise into feature representations, varying across different timesteps. The trained diffusion model predicts the added noise in these noisy features based on text instruction conditioning. This paper concentrates on latent diffusion models (Rombach et al., 2022), which learn the diffusion process in the latent

space rather than the image space. Specifically, we employ Stable Diffusion XL v1 (Podell et al., 2023) for all our experiments.

**LoRA Combination**

Low-Rank Adaptation (LoRA) (Hu et al., 2021) is a method for efficient adaptation of Large Language and Vision Models to a new downstream task. The key concept of LoRA is integrating additional trainable low-rank matrices within the neural network. Specifically, for a weight matrix $W \in \mathbb{R}^{n \times m}$ in the pre-trained model, the update of W after applying LoRA is formulated as $W^{'} = W + \Delta W$, where $\Delta W = BA$. Here, $B \in \mathbb{R}^{n \times r}$ and $A \in \mathbb{R}^{r \times m}$. The low-rank factor $r$ satisfies $r << min(n, m)$. During training, only A and B are updated to find suitable $\Delta W = BA$, while keeping $W$ constant. Due to its efficiency, LoRA is widely used for fine-tuning open-sourced diffusion models (Podell et al., 2023).

To generate images containing several distinct characters or styles, a series of LoRA combination methods are proposed, one of which is LoRA Merge. The concept of LoRA Merge is realized by linearly combining multiple LoRAs to synthesize a unified LoRA, subsequently plugged into the diffusion model. Formally, when introducing $n$ different LoRAs, the update of $W$ are as follows.

$$W^{'} = W + \sum_{k=1}^{n} w_i \times B_k A_k, \tag{1}$$

where $w_i$ stands for the combination coefficient. Other LoRA combination methods either require additional gradient computations to update to $w_i$ Shah et al. (2023) or avoid tuning $w_i$ by altering the forward pass of diffusion models. Therefore, these methods require more time (**around several hours**) and they may still under-perform than naive adjustment of the combination coefficient. On the contrary, manual adjustment enjoys fast inference speed (**around several seconds**), but it requires tens or hundreds of attempts, especially when the number of LoRAs increases. This paper investigates the potential of directly adjusting combination coefficients for LoRA combination by harnessing the in-context learning ability of MLLMs, which, to our knowledge, has not been explored before.

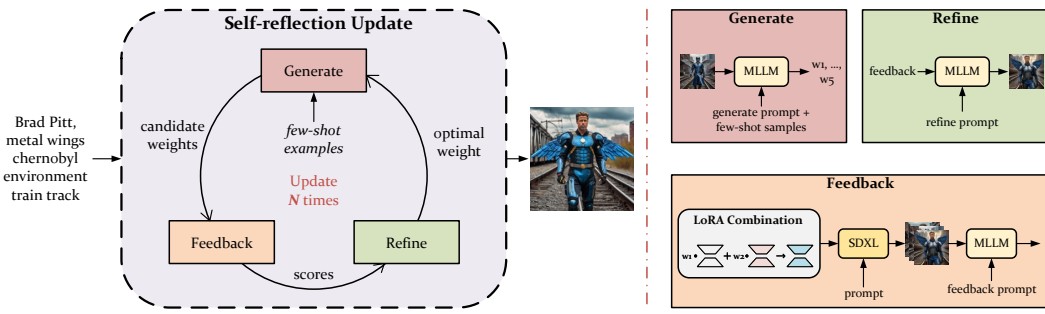

Figure 2: **GPT4LoRA Overview**. GPT4LoRA mainly consists of three steps: *Generate*, *Feedback*, and *Refine*. These steps formulate an iterative refinement procedure, following the logic of self-reflection.

### 3.2 ITERATIVE REFINEMENT WITH GPT4LORA

Given a user-defined textual prompt and several LoRA models as inputs, GPT4LoRA generates the candidate weights, provides feedback on the candidate weight, and refines the candidate weight according to the feedback. GPT4LoRA iterates among these steps until the iterative refinement procedure ends. GPT4LoRA relies on a suitable multimodal large language model and three prompts (for generate, feedback, and refine), and does not require training. The overview of GPT4LoRA is shown in Figure 2 and Algorithm 1. Next, we describe GPT4LoRA in more detail.

#### 3.2.1 FEW-SHOT SAMPLE SELECTION

Unlike previous methods Lee et al. (2024); Xu et al. (2023) where annotations, e.g., cropping coordinates, are available, the absence of standardized benchmarks in LoRA combination areas hinders

the selection of few-shot samples, as the performance is highly sensitive to the quality of the chosen few-shot samples (Liu et al., 2021; Lu et al., 2021). To this end, we propose a few-shot sample selection paradigm for LoRA combination to better prompt MLLMs. Specifically, when combining multiple LoRA models, given an input text description, we can generate a set of images based on all possible combinations of coefficients. We then calculate the text-alignment scores of the generated images w.r.t. the given text description and rank these images according to their scores. Directly selecting generation samples with the highest text-alignment scores may result in unbalanced combination coefficients. This phenomenon primarily arises from explicit information leakage, where certain LoRA models contain trigger phrases that prompt the pre-trained text-to-image model to generate the desired image even without incorporating the corresponding LoRA model. As pointed out in the previous study, LoRA combination with unbalanced weights will destabilize the combination process (Huang et al., 2023). To overcome this issue, we simply filter out images with a minimum score of less than a pre-defined threshold. After obtaining the filtered samples, we selected the samples with the top-5 highest text-similarity scores, i.e, $\{(\hat{i}_1, \hat{w}_1), ..., (\hat{i}_5, \hat{w}_5)\}$, to formulate the few-shot samples.

---

**Algorithm 1:** GPT4LoRA

---

**Input:** textual prompt $t$, LoRA models $\{L_k, t_k\}_{k=1}^k$
**Prerequisite:** iterations $N$, MLLM $M$, SDXL $G$, generate prompt $p_{\text{gen}}$, feedback prompt $p_{\text{fb}}$, refine prompt $p_{\text{re}}$, few-shot samples $s$, combination coefficient $w$, number of candidate weights $M$, current iteration $r$
**Output:** Image $I$
$r \leftarrow 0$;
**while** $r < N$ **do**
$\quad w_1, ..., w_M \leftarrow M(p_{\text{gen}}(s), G(t, \{L_k\}_{k=1}^k, w), [G(t_i, L_i)|i \in 1, ..., k])$   // generate;
$\quad \text{fb}_r \leftarrow M(p_{\text{fb}}, [G(t, \{L_k\}_{k=1}^k, w_i)|i \in 1, ..., M], [G(t_i, L_i)|i \in 1, ..., k])$ // feedback;
$\quad w \leftarrow M(p_{\text{re}}, \text{fb}_r, [G(t, \{L_k\}_{k=1}^k, w_i)|i \in 1, ..., M])$         // refine;
$\quad \hat{I}_r \leftarrow G(t, \{L_k\}_{k=1}^k, w)$;
$\quad r \leftarrow r + 1$;
**end**
$I \leftarrow M(p_{\text{re}}, I_1, ..., I_N)$;
**Return:** $I$

---

### 3.2.2 Optimization LoRA combination via self-reflection

**Generate** Given LoRA models $\{L_k, t_k\}_{k=1}^k$, a text prompt $t$, a generate prompt $p_{gen}$, few-shot samples $s$, and a MLLM $M$, GPT4LoRA generates several candidate combination coefficients (set to 5 by default).

$$w_1, ..., w_5 \leftarrow M(p_{\text{gen}}(s), G(t, \{L_k\}_{k=1}^k, w), [G(t_i, L_i)|i \in 1, ..., k]). \quad (2)$$

Here, $p_{gen}$ is a task-specific few-shot prompt (or instruction) for generation and the few-shot samples contain input-output pairs $< (t, L), w >$ for LoRA combination.

**Feedback** Without explicit supervision, MLLM lacks a deep understanding of the context of the LoRA combination task, such as the understanding of certain styles at a fine-grained level. Consequently, it may produce nonsensical outputs even with good ICL samples. Empirically, we observe that the initial combination coefficient candidates generated by the GPT-4o lack diversity and sometimes fail to make sense. Previous study (Yang et al., 2024a) has shown that large language models can optimize the output by iteratively incorporating feedback. To this end, GPT4LoRA utilizes GPT-4o as a qualified evaluator to provide fruitful feedback. Given separate LoRA models' information, intermediate images that are generated given the candidate combination coefficients, and a task-specific prompt $p_{fb}$ for generating feedback, GPT4LoRA uses the same model $M$ to provide feedback fb on its own output:

$$\text{fb} \leftarrow M(p_{\text{fb}}, [G(t, \{L_k\}_{k=1}^k, w_i)|i \in 1, ..., M], [G(t_i, L_i)|i \in 1, ..., k]). \quad (3)$$

Intuitively, the feedback may contain constructive information on how the input LoRA models behave and interact with each other.

**Refine** Finally, GPT4LoRA uses $M$ to refine its last output and select the optimal combination coefficient, given its own feedback:

$$w \leftarrow M(p_{\text{re}}, \text{fb}, [G(t, \{L_k\}_{k=1}^k, w_i)|i \in 1, ..., M]). \tag{4}$$

**Iterating GPT4LoRA**

GPT4LoRA alternates among *generate*, *feedback* and *refine* steps until the iteration ends. This iterative process is repeated $N$ times, and the top output is selected as the final result. Details of the prompt design are shown in the supplementary material.

## 4 EXPERIMENTS

### 4.1 EXPERIMENTAL SETUP

**Implementation Details**

In our experiments, we utilize Stable Diffusion XL (Podell et al., 2023) as the backbone model. For a thorough evaluation, we use two specific checkpoints: "SDXL-vae-fix" for realistic images and "Animagine-xl-3.1" for anime images. For generating realistic images, we configure the model with 50 denoising steps, and a guidance scale of 7, and employ the Euler scheduler for the diffusion process. The image resolution is set to 1024x1024 pixels to enhance quality. In contrast, for anime-style images, we adjust the settings to 30 denoising steps, a guidance scale of 6, and use the Euler Ancestral scheduler, maintaining the same image resolution of 1024x1024 pixels. For both types of images, we set the number of total updates to 5 and the number of candidate weights to 5. To ensure the robustness of our results, we generate images using three different random seeds. All reported results represent the average evaluation scores across these three trials.

**Inference Details**

We have selected two distinct subsets of LoRAs that represent realistic and anime styles. Each subset includes a diverse mix of elements: characters, clothing, styles, and backgrounds. Altogether, these subsets form a collection of 24 LoRA models in total. In constructing inference sets, we adhere to a key principle: each set must include one character LoRA and avoid duplicating element categories to prevent conflicts. Consequently, our evaluation comprises 105 distinct composition sets. Trigger words, i.e., key features, are manually annotated. These trigger words serve as input prompts for the text-to-image models to generate images and as reference points for subsequent evaluation using GPT-4o. Detailed descriptions of each LoRA are provided in the Appendix. The main experiments are performed to fulfill the combination of three LoRA models, one for character, one for clothing, and the other one for style or background. LoRA Merge, LoRA Switch Zhong et al. (2024), and LoRA Composite Zhong et al. (2024) are chosen as the baseline methods for their ability to combine multiple LoRA models. We also provide the experimental results of combining two LoRA models (including ZipLoRA (Shah et al., 2023)) in the supplementary material.

**Evaluation Metrics** Following DreamBooth (Ruiz et al., 2023), we provide comparisons of image-alignment and text-alignment scores. Furthermore, we also leverage GPT-4o's capabilities to serve as an evaluator for LoRA combination-based image generation. This MLLM-based evaluation involves scoring the performance of two comparative results across two dimensions, as well as determining the winner based on these scores

### 4.2 COMPARATIVE EVALUATION WITH GPT-4O

While existing quantitative metrics, e.g., image-alignment and text-alignment scores, can calculate the alignment between text and images (Shah et al., 2023; Zhong et al., 2024), they fail to capture subtle stylistic details and are intertwined with the semantic properties of images, including their overall content. Recent studies (Zhong et al., 2024; Zhang et al., 2023c) demonstrate the efficiency of MLLMs in evaluating various multimodal tasks, underscoring their potential in evaluating image generation tasks. As a comprehensive evaluation, we leverage GPT-4o's ability to serve as a discriminator to evaluate generated images in two dimensions: composition quality and image quality with the former evaluating local details restoration and the latter evaluating from a rather global perspective. We present an example in Table 1. Besides, for a more fair comparison, we repeat the

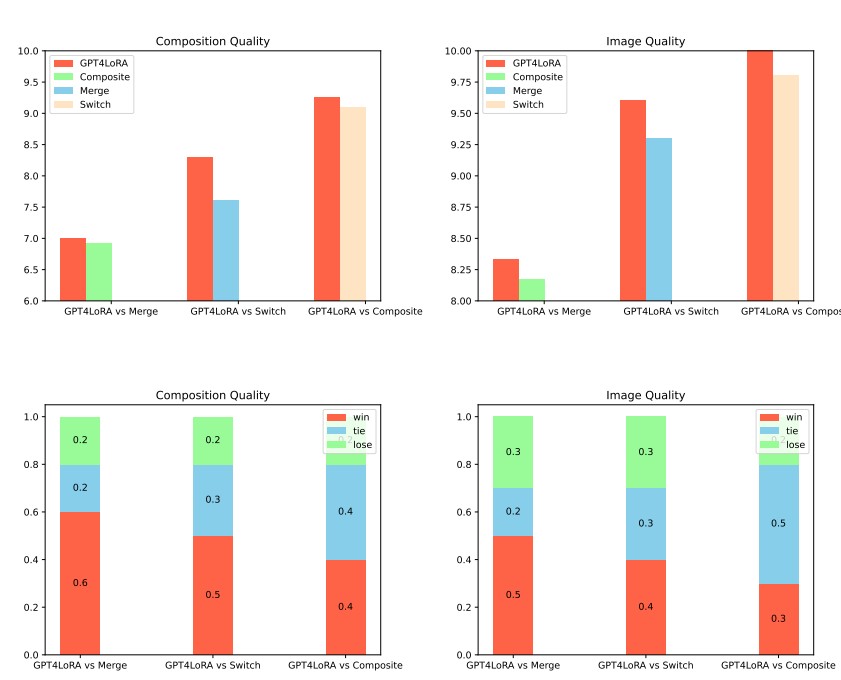

Figure 3: Results of comparative evaluation using GPT-4o.

MLLM-based evaluation 10 times to calculate the specific win rates and provide specific scores and win rates in Figure 3. It can be observed from Figure 3 that our proposed GPT4LoRA consistently outperforms existing methods across both composition quality and image quality.

Table 1: Example of GPT-4o-based evaluation. The evaluation prompt and result are in a simplified version.

---

**Evaluation Prompt**

I need assistance in comparatively evaluating two text-to-image models based on their ability to compose different elements into a single image. The key elements are:
1. Character: ganyu, black gloves;
2. Clothing: black legwear, hair ribbon, dress, short sleeves, frills apron, puffy short sleeves;
3. Style: lineart, traditional media, sketch, monochrome, greyscale;
Please help me rate based on composition and image quality:

**Evaluation Results from GPT-4o**

...
**For Image 2**:
Composition Quality:
Dress: *Present but colored.*
Short sleeves: *Present with puffy detailing.*
Monochrome: *No, has blue tones.*
Image Quality:
*Consistent but lacks detailed variation.*
*Dress color can be considered a minor flaw affecting coherence.*
...
Scores:
- Image 1: Composition Quality: 10/10, Image Quality: 10/10
- Image 2: Composition Quality: 5/10, Image Quality: 8.5/10

---

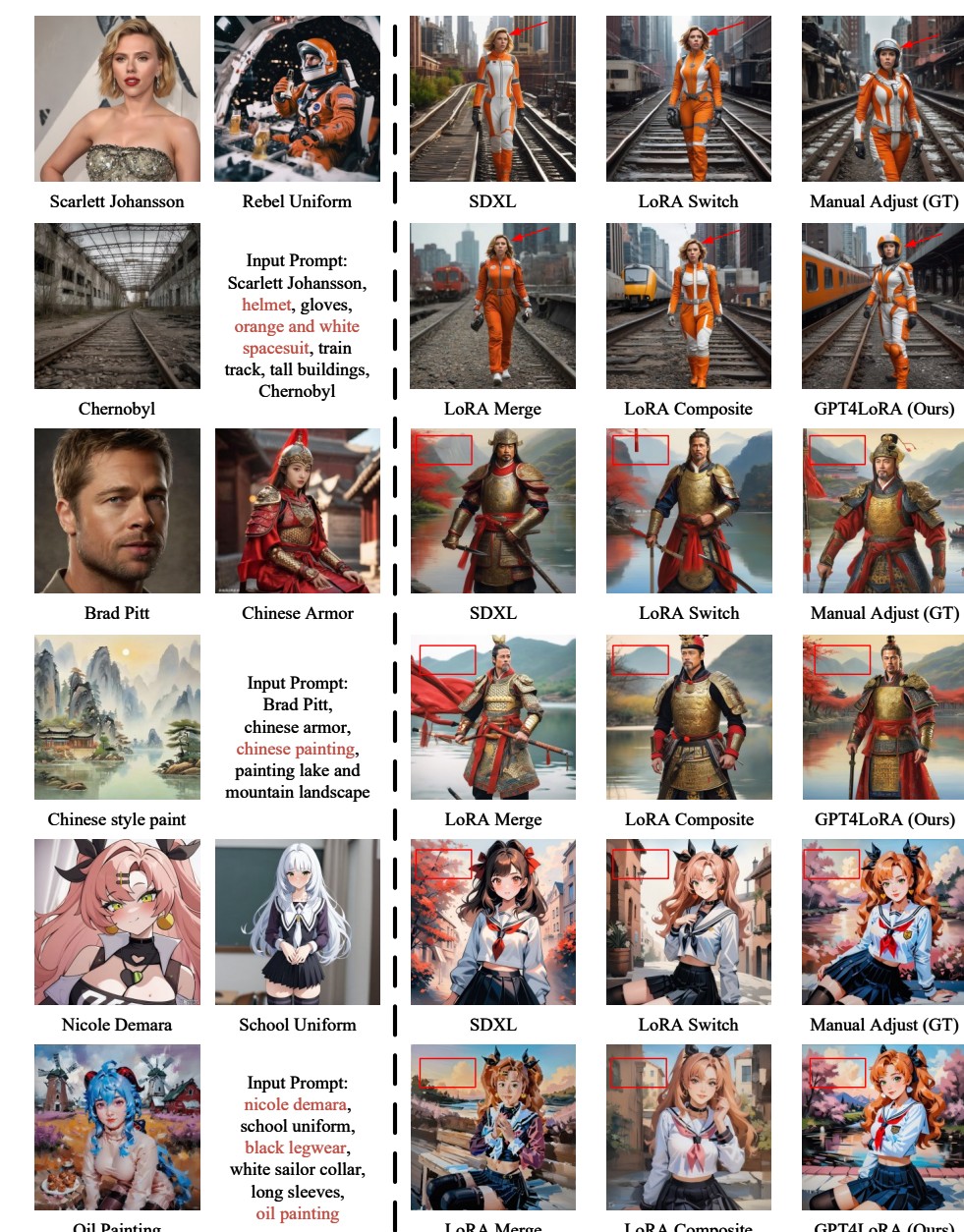

Figure 4: Visual Comparisons between GPT4LoRA and other LoRA combination methods. Key areas are marked with red boxes or arrows.

## 4.3 VISUAL COMPARISON AND QUANTITATIVE RESULTS

We use CLIP-I scores of image embeddings of output and the style reference for image-alignment, as well as CLIP-T embeddings of the output and the text prompt for text-alignment. We evaluate realistic and anime subsets respectively, the quantitative results are presented in Table 2. It can be observed that GPT4LoRA surpasses current methods in image and text alignment, indicating its proficiency in maintaining text-to-image generation capabilities while effectively expressing the specified style and subject outlined in the text prompt. Besides, we present the visual comparison between GPT4LoRA and other methods in Figure 4, where we also include manual adjust to comparison. It can be observed that GPT4LoRA not only generates objects that are strictly coherent to prompt but also seamlessly integrates different styles.

Table 2: Quantitative Results between GPT4LoRA and other LoRA combination methods.

|  | LoRA Merge | LoRA Switch | LoRA Composite | GPT4LoRA |
|---|---|---|---|---|
| Realistic CLIP-I | 0.6026 | 0.6117 | 0.6109 | 0.6191 |
| Realistic CLIP-T | 0.3429 | 0.3387 | 0.3501 | 0.3561 |
| Anime CLIP-I | 0.6767 | 0.6713 | 0.6789 | 0.6827 |
| Anime CLIP-T | 0.3023 | 0.2869 | 0.3011 | 0.3082 |

### 4.4 ANALYSIS

To better enhance the understanding of the proposed GPT4LoRA, we further investigate the following critical questions:

#### 4.4.1 DOES GPT-4O KNOW HOW THE DIRECTION AND AMOUNT OF TUNING COMBINATION COEFFICIENTS?

To explore this, we perform the following ablation experiments. Three LoRA models were given to compose by ignoring style-LoRA's trigger words in the input prompt. We present the visual comparison in Figure 5. It can be observed that GPT4LoRA generates an impressive image that is coherent with the input prompt and does not corrupt the image with irrelevant LoRA.

Figure 5: Ablation study on ignoring some trigger words.

#### 4.4.2 TO WHAT EXTENT DO THE FEW-SHOT SAMPLES INFLUENCE THE FINAL PERFORMANCE?

To explore this, we perform the following ablation experiments. Given three LoRA models to compose, we ignore the few-shot sample information during prompting GPT-4o. We present the quantitative comparison w.r.t text-alignemnt and image-alignment in Table 3. Without few-shot samples, GPT-4o tends to generate nonsensical and repetitive responses Lee et al. (2024), which fails to grasp the implicit interaction among different LoRA models and poses inferior performance in both text- and image-alignment.

Table 3: Ablation studies on the impact of few-shot samples.

|  | Realistic CLIP-I | Realistic CLIP-T | Anime CLIP-I | Anime CLIP-T |
|---|---|---|---|---|
| w/o few-shot samples | 0.5994 | 0.3218 | 0.6265 | 0.2745 |
| w/ few-shot samples | 0.6191 | 0.3561 | 0.6827 | 0.3082 |

## 5 CONCLUSION

This paper presents GPT4LoRA, the first exploration of utilizing of self-reflection mechanism in MLLMs for LoRA combination. By a carefully designed paradigm for few-shot sample selection, which guides the self-reflection mechanism of the MLLM during the iterative process, the proposed GPT4LoRA does not require annotated data or manually designed rules, instead relying on few-shot samples and specifically tailored prompts for generating, evaluating, and refining LoRA combinations. Extensive experiments conducted on a benchmark of widely-used LoRA models demonstrate that GPT4LoRA outperforms existing methods in both quantitative and qualitative evaluations. By eliminating reliance on unreliable automatic scoring systems and harnessing MLLM-based self-reflection, GPT4LoRA establishes a new standard for efficient and high-quality LoRA composition in generative image models.

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

## A    APPENDIX

You may include other additional sections here.

