# 1 DETAILS OF EACH LoRA

Table 1: Detailed descriptions of each LoRA.

| LoRA | Category | Trigger Words | Source |
|------|----------|---------------|--------|
| *Anime-style* | | | |
| Ganyu | character | 1girl, ganyu, looking at viewer, hair flower, black gloves, blush | website |
| Nicole Demara | character | 1girl, nicole demara, hair ribbon, crop top | website |
| Yor Forger | character | 1girl, yor briar, closed mouth, hand up, smile, looking at viewer, earrings | website |
| Cosplay | clothing | alicecos, black legwear, red hair ribbon, blue dress, white short sleeves | website |
| School Uniform | clothing | school uniform, serafuku, black legwear, white sailor collar, long sleeves | website |
| Coffee Dress | clothing | starbuni, emblem, green apron, starbucks, cap | website |
| Linear Art | style | lineart, traditional media, sketch, monochrome, greyscale | website |
| Pixel Art | style | pixel art | website |
| Oil Painting | style | oil painting, oil painting style | website |
| Shoelocker | background | getabako, scenery, window, locker, school, indoors | website |
| Kitchen | background | indoors, kitchen, day, windows, kitchen interior | website |
| *Realistic-style* | | | |
| Gal Gadot | character | portrait photography of Gal Gadot, a beautiful woman, full-body | website |
| Jennifer Lawrence | character | close-up portrait photography of Jennifer Lawrence, full-body | website |
| Scarlett Johansson | character | professional portrait photography of Scarlett Johansson, full-body | website |
| Brad Pitt | character | closeup portrait of Brad Pitt, 3/4 portrait | website |
| Hanfu | clothing | szhf dress, white beizi, wide sleeve, green belts, white pleated skirt | website |
| Chinese Armor | clothing | breathtaking Qinjia, a 40 year old man, red Chinese armor, black shoulder armor, golden helmet | website |
| Hades | clothing | Cloud Burst Blue hdsrmr, giant metal wings, helmet, mask | website |
| Rebel Costume | clothing | helmet, gloves, orange and white spacesuit | website |
| Chernobyl | background | a train track running through a city with tall buildings, Chern4byl environment | website |
| Japan | background | japan, scenery, outdoors, road, power lines, building | website |
| Hotel | background | ryokan, table, indoors, scenery | website |
| Chinese Style | style | guofeng, chinese style, traditional Chinese painting, painting lake and mountain landscape, mountain reflection on lake surface | website |
| Oil Paint | style | bichu, oil painting, ancient magical ruins, deep forest, overgrown with vines | website |

## 2 VISUAL COMPARISON

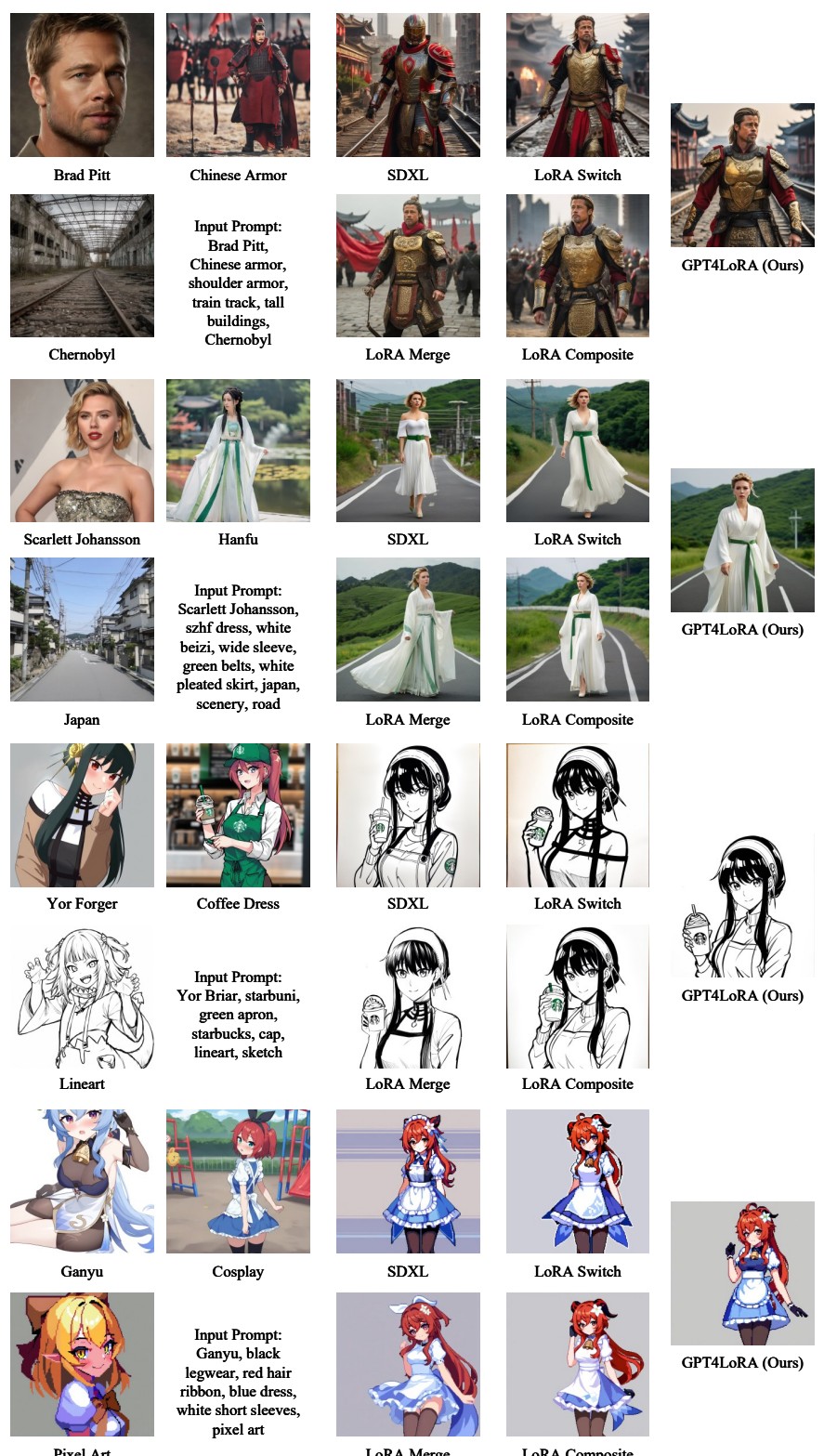

Figure 1: Visual Comparison between GPT4LoRA and other methods.

## 3 PROMPT DESIGN

**Generation Prompt**

Your task is to iteratively estimate the optimal coefficients for composing three LoRA modules, ensuring that the merged image represents the styles and contents of all LoRAs well. Here's a detailed approach: 1. *Analyze Individual LoRA Images*: Start by carefully examining the content and style of the demo images produced by each LoRA module. Note key characteristics that define each module's unique style and content; 2. *Analyze the Merged Image*: Compare the merged image with the individual LoRA demo images. Then determine which styles and contents are underrepresented or overrepresented in the merged image.; 3. *Adjust Coefficients*: Decide on necessary coefficient adjustments. Remember the goal is to achieve an equal representation of styles and contents from all three LoRAs.; 4. *Format of the Answer*: Once you determine the optimal coefficients, provide them in the specified format. We provide several examples here.; {LoRA sample} $(w_1^1, ..., w_1^5)$ {Query sample}.

**Feedback Prompt**

I need assistance in comparatively evaluating one text-to-image model based on its ability to compose different elements into a single image. The key elements are: 1. Character: XXX; 2. Clothing: XXX; 3. Style: XXX; Please help me rate based on composition and image quality.

**Refine Prompt** Please evaluate the following five merged images and select the one that best incorporates the contents and styles demonstrated in the provided demo images. Your response should be formatted as follows: The merged image at count XXX has the best visual effect.