# OpenReview forum: "GPT4LoRA: Optimizing LoRA Combination via MLLM Self-Reflection"
_ICLR.cc/2025/Conference — Submitted to ICLR 2025_

### Official Review · Reviewer_oSnY · 2024-10-30

**Soundness:** 1
**Presentation:** 1
**Contribution:** 2
**Rating:** 3
**Confidence:** 4

**Summary:**

This paper explores using a self-reflection mechanism for LoRA combination. Specifically, it follows a "generation, feedback, and refine" paradigm, where MLLMs are leveraged to provide feedback to the previous round of generation with a set of carefully selected demonstrations as prompts. The process is iterated for several rounds until the best weight of the combination is achieved. Experiments on   a newly proposed benchmark by the authors verified the effectiveness of the methods, in terms of GPT-4o evaluation and CLIP scores.

**Strengths:**

1. This paper researches an interesting problem of composing multiple LoRAs for image customization.
2. The proposed framework of leveraging MLLMs to provide feedback in the process is intuitive.

**Weaknesses:**

1. The paper presentation needs significant improvement. Specifically, the top two figures of Figure 3 contain inconsistent information. Labels in the x-axis do not align with the legends. Legend in the bottom two figures of Figure 3 block the third bars. The authors should also be cautious of citation format, e.g., lines 305-306, line 215.
2. The results in Table 2 are not significant enough compared with LoRA-Composite in terms of CLIP scores. A significant test would be helpful.
3. Results in Table 3 show that without few-shot demonstrations, the performance is seriously downgraded, even inferior to LoRA merge. This makes me doubt the actual effectiveness brought by the "generate, feedback, and refine" pipeline.
4. Some details are missing for experiments. See questions below.

**Questions:**

1. In lines 228-229, have you experimented with other choices of $k$ instead of $5$? How is the number $5$ determined?
2. The authors should provide implementation details of LoRA composite and LoRA switch. In their original paper, the backbone diffusion model and image resolution are quite different from GPT4LoRA. The details could help readers understand and to ensure the fairness of comparison.
3. If I understand it correctly, the MLLM used is GPT-4o and the same model is used for evaluation. I wonder if there exists any biases towards the evaluation.

---

### Official Review · Reviewer_ipKv · 2024-11-04

**Soundness:** 2
**Presentation:** 2
**Contribution:** 2
**Rating:** 3
**Confidence:** 4

**Summary:**

This paper introduces GPT4LoRA, a method designed to optimize the combination of LoRA models for generative image synthesis using the self-reflection capabilities of MLLMs. GPT4LoRA is a training-free framework with three-step process: Generate, Feedback, and Refine. This iterative framework leverages MLLMs to adjust coefficient weights without modifying the underlying model architecture​.

The paper evaluates GPT4LoRA against existing methods using a combination of quantitative metrics and GPT-4o-based assessments. The experiments focus on maintaining alignment between generated images and textual prompts across realistic and anime styles. The authors claim that GPT4LoRA achieves improvements over baseline methods in both composition quality and image coherence.

**Strengths:**

1. The authors integrate MLLMs and diffusion models, proposing a new approach to optimize LoRA composition.
2. They construct a testbed with 24 LoRA models based on SDXL.
3. Experimentally, GPT4LoRA provides some improvement over existing LoRA composition methods.

**Weaknesses:**

**Major Issues:**

1. **Efficiency Concerns:** GPT4LoRA’s efficiency is a significant concern. To combine multiple LoRAs into a single image, the proposed framework requires GPT-4o to first generate coefficients based on few-shot samples, then use SDXL to generate multiple images, followed by GPT-4o generating feedback to refine textual prompts and coefficients. Additionally, this process is iterated $N$ times. In sum, generating a single image can potentially require prompting GPT-4o over a dozen times and generating dozens of candidate images with SDXL. Given the relatively marginal improvements seen in qualitative and quantitative experiments, this substantial increase in computational cost may be difficult to justify.

2. **Limited Experiments:** Compared to LoRA Switch and LoRA Composite, GPT4LoRA's experiments are relatively limited. Specifically, this paper evaluates only 3-LoRA combinations with 105 composition sets, while previous works have explored 2-5 LoRA combinations and included more composition sets. Prior studies also incorporated human evaluation, which this work lacks. Analytical experiments are also sparse. For example, only an ablation on few-shot samples is provided, while there is no analysis of each step or the number of iterations in the framework. Since each step and additional iteration increases significant computational cost, analyzing these aspects would be valuable.

3. **Inconsistent Claims:** The authors' motivations appear unsupported due to inconsistencies. In the Introduction, the authors state that prior work suffers from (1) being "computationally costly and impractical when a large number of LoRA models are involved" (Lines 58-59). However, as discussed above, GPT4LoRA’s computation demands are notably higher than existing methods. They also claim (2) "A fundamental limitation of these methods lies in the subjectivity and unreliability of the evaluation process for image quality" (Lines 59-69), yet the MLLM self-reflection and GPT-based evaluation used here are established in prior work. Thus, the motivations behind GPT4LoRA seem unsubstantiated.

**Minor Issues:**

1. **Overlap with Prior Work:** Parts of this paper closely resemble the previous study [1]. For example, the entire "Diffusion Models" paragraph in 3.1, as well as the first two paragraphs in "LoRA Combination," show only minor paraphrasing of prior work. Besides, Table 1 is almost the same as Table 1 in [1] (but there is no mention of evaluation criteria and format requirements). In other words, the GPT-based evaluation approach (comparative evaluation, two evaluation dimensions, point-wise scoring, win rate, and evaluation prompts) is identical to [1] but lacks explicit citation and description.

2. **Lack of Detail:** Several key details are missing. See Questions 1-4 below.

3. **Unexplained Results:** Certain results seem confusing and lack analysis. See Questions 5-6 below.

**Typos:**

1. The legend and x-axis in the point-wise part of Figure 3 do not match.

**References:**

[1] Zhong et al. Multi-LoRA Composition for Image Generation.

**Questions:**

1. Since [1] provides a testbed with 480 composition sets, why did the authors create a new benchmark with only 105 composition sets? The new benchmark omits the "object" category and is otherwise identical.

2. Why are there 105 composition sets for 24 LoRAs? Based on supplementary material Table 1, anime-style compositions should yield 3&times;3&times;(3+2)=45 sets, and realistic-style should yield 4&times;4&times;(3+2)=80, for a total of 125 sets.

3. Given that GPT models exhibit position bias when used as evaluators, did the authors average scores by switching the image positions in comparative evaluations?

4. The paper states, "We also provide the experimental results of combining two LoRA models (including ZipLoRA (Shah et al., 2023)) in the supplementary material," (Lines 307-308) but the supplementary material appears to lack any additional experiments.

5. In Figure 3, why do point-wise evaluation scores vary significantly when switching baseline models? For example, GPT4LoRA's score is around 7.0 when compared with LoRA Merge, but exceeds 9 when compared with LoRA Composite.

6. If there are 105 composition sets, and MLLM-based evaluation is repeated 10 times with 3 random seeds per image, this should result in over 3K comparative evaluations. Why do the win rates in Figure 3 show only a single decimal place?

If I have misunderstood any of these weaknesses or questions, please feel free to correct me.

---

### Official Review · Reviewer_HyQA · 2024-11-04

**Soundness:** 3
**Presentation:** 3
**Contribution:** 2
**Rating:** 5
**Confidence:** 3

**Summary:**

This paper introduces GPT4LoRA, a method that leverages the self-reflection capabilities of multimodal large language models (MLLMs) to enhance Low-Rank Adaptation (LoRA) model combinations for generative tasks. Traditional LoRA combination approaches often require additional fine-tuning or changes to model architecture, which GPT4LoRA addresses through a training-free, three-step process: Generate, Feedback, and Refine. Extensive experiments conducted on the realistic and anime-style datasets show that GPT4LoRA outperforms existing methods in both quantitative and qualitative evaluations.

**Strengths:**

(1)GPT4LoRA’s use of MLLM self-reflection introduces a new paradigm for training-free LoRA model combinations.

(2)Extensive experiments demonstrate superior performance compared to baseline methods.

**Weaknesses:**

(1) The evaluation benchmark used in this paper is unclear. The paper mentions “Extensive experiments conducted on a benchmark of widely-used LoRA models” in lines 83 and 482, but lacks citations, leaving it unclear which text-to-image evaluation dataset is used.

(2) The paper lacks comparisons with comparable methods, such as ZipLoRA and LoRA Composite. ZipLoRA used DreamBooth and StyleDrop as evaluation datasets—could authors evaluate the GPT4LoRA on these datasets and choose ZipLoRA as the strong baseline model?

(3) The method's reliance on the self-reflection capabilities of multimodal large language models (MLLMs) like GPT-4 may result in variable outcomes depending on the MLLM's quality and adaptability, potentially limiting robustness across different models.

(4) While few-shot sample selection is critical to GPT4LoRA's success, details about this process are sparse, and the choice of demonstration samples significantly impacts performance, which may make it challenging for other researchers to reproduce the results effectively.

(5) A minor issue: some references in the paper are improperly formatted. For example, in line 144, "Reflexion Shinn et al. (2024) converts," and in line 214, "Unlike previous methods Lee et al. (2024); Xu et al. (2023)."

(6) Some results in the paper show limited improvements; it’s recommended that the authors conduct significance tests to analyze these improvements.

**Questions:**

See the Weaknesses.

---

### Official Review · Reviewer_oaJ1 · 2024-11-05

**Soundness:** 2
**Presentation:** 2
**Contribution:** 2
**Rating:** 3
**Confidence:** 3

**Summary:**

The paper proposes a training free method to combine multiple LoRA trained weights to generate better aligned images with text prompts in Text2Image setup. Building on top of LORA-merge, where multiple LORA trained weights are linearly combined, this paper utilizes a multimodal LLM to directly generate the combination weights. Furthermore, the approach utilizes the same multimodal LLM to refine its generations using feedback and self refinement.

**Strengths:**

- The paper presents a training-free method of generating linear combination of merging multiple LORA trained weights, utilizing GPT4o.
- The paper shows qualitatively that these approaches offer a better and cheaper method to controlling the characteristics of the generated image.

**Weaknesses:**

- The paper is low on contributions - the method, while interesting, probably doesn't warrant a full paper - its better suited for a blog post.
- The paper is weak on quantitative results - Table 2 results does not appear statistically significant.
- The paper lacks analysis on the GPT4o outputs of the linear combinations.
- Experiments only conducted on closed source GPT4o, it is unclear if this kind of approach works for open source models, thereby limiting the applicability.

**Questions:**

- What happens when the number of LORA weights is increased from three?

---

### Meta-Review · Area_Chair_5pAV · 2024-12-18

**Metareview:**

**Summary:**

This paper introduces GPT4LoRA, a training-free method to optimize the combination of LoRA models for text-to-image generative tasks. GPT4LoRA leverages the self-reflection capabilities of multimodal large language models in a three-step iterative framework: Generate, Feedback, and Refine. By building on LoRA-merge, it uses MLLMs to dynamically generate and refine combination weights without modifying the underlying architecture. Experiments on realistic and anime-style datasets demonstrate that GPT4LoRA improves alignment between text prompts and generated images, outperforming baseline methods in composition quality and image coherence through both quantitative metrics and GPT-4-based assessments.

**Weaknesses:**

* The evaluation benchmark is not well-defined, with no citations or clarity on the specific text-to-image datasets used.
* Few-shot sample selection is critical but inadequately detailed, making reproducibility challenging.
* Marginal improvements in results lack statistical significance tests to validate their relevance.
* Evaluations are restricted to 3-LoRA combinations with fewer composition sets compared to prior works, and no human evaluation or thorough analysis of framework steps and iterations is provided.

**Additional Comments On Reviewer Discussion:**

No Rebuttal Discussion.

**Decision:** This paper received two rejections but the authors chose not to participate the rebuttal process. Therefore, I recommend rejection.

---

### Decision · Program_Chairs · 2025-01-22

Reject